# Nucleoid remodeling during environmental adaptation is regulated by HU-dependent DNA bundling

Soumya G. Remesh[1,2,4], Subhash C. Verma [1,4], Jian-Hua Chen[2,3], Axel A. Ekman[2,3], Carolyn A. Larabell[2,3], Sankar Adhya[1] & Michal Hammel [2✉]

Bacterial nucleoid remodeling dependent on conserved histone-like protein, HU is one of the determining factors in global gene regulation. By imaging of near-native, unlabeled *E. coli* cells by soft X-ray tomography, we show that HU remodels nucleoids by promoting the formation of a dense condensed core surrounded by less condensed isolated domains. Nucleoid remodeling during cell growth and environmental adaptation correlate with pH and ionic strength controlled molecular switch that regulated HUαα dependent intermolecular DNA bundling. Through crystallographic and solution-based studies we show that these effects mechanistically rely on HUαα promiscuity in forming multiple electrostatically driven multimerization interfaces. Changes in DNA bundling consequently affects gene expression globally, likely by constrained DNA supercoiling. Taken together our findings unveil a critical function of HU–DNA interaction in nucleoid remodeling that may serve as a general microbial mechanism for transcriptional regulation to synchronize genetic responses during the cell cycle and adapt to changing environments.

[1] Laboratory of Molecular Biology, Center for Cancer Research, National Cancer Institute, Bethesda, MD 20892, USA. [2] Molecular Biophysics and Integrated Bioimaging, Lawrence Berkeley National Laboratory, Berkeley, CA 94720, USA. [3] Department of Anatomy, University of California San Francisco, San Francisco, CA 94158, USA. [4]These authors contributed equally: Soumya G. Remesh, Subhash C. Verma. ✉email: mhammel@lbl.gov

Nucleoid remodeling facilitated by DNA organization activities of nucleoid-associated proteins (NAPs) is one of the determining factors of global gene regulation[1–4]. HU is among the most conserved and abundant NAPs in eubacteria and has major role in the nucleoid structure[5] (Supplementary Figure 1a). Mutations or the deletion of HU transform the *Escherichia coli* nucleoid to different forms and alter transcription program[6–8]. HU causes gene expression changes by modulating the 3D arrangement of DNA at different length scales within the nucleoid; by facilitating DNA looping in a promoter region, trapping free supercoils, indirectly altering supercoiling through DNA topoisomerases, or perhaps by promoting long-range DNA–DNA contacts[6,7,9,10]. As opposed to other NAPs (H-NS, Fis, and IHF) that are specifically localized within the nucleoid due to DNA sequence-specific binding, HU is largely scattered throughout the whole nucleoid owing to its non-sequence-specific DNA binding[11,12]. Although HU binds DNA regardless of the sequence it exhibits two DNA-binding modes. It binds with low-affinity to linear DNA through a phosphate lock mechanism but binds damaged (kinked) or cruciform DNA specifically with high affinity through its long β-ribbon arms[13,14]. High-affinity sites are presumably much fewer in the *E. coli* chromosome and HU mostly interacts with chromosomal DNA through low-affinity binding. In *E. coli*, HU exists as homo- or heterodimers of HUα and HUβ subunits. Owing to differential expression and stability of the two subunits during *E. coli* growth cycle, HUαα predominates in the growth phase, whereas HUαβ in the stationary phase[15], pointing to distinct roles of HUαα and HUαβ in nucleoid architecture and gene expression during growth and stasis, respectively. We have previously shown that the non-sequence-specific DNA-binding mode of HU causes bundling of independent DNA strands through HUαα–HUαα multimerization that is maintained by a zipper-like network of hydrogen bonds of highly ionized amino acids[7]. However, whether HU-mediated DNA bundling depends on environmental changes and growth conditions remained unclear.

In this study, we provide a holistic view of the molecular connections between HU–DNA interactions and the nucleoid architecture as it correlates to global regulation of bacterial gene expression during growth, stasis and under stress. Using an X-ray based imaging technique called soft X-ray tomography (SXT)[16], we characterized the higher-order *E. coli* nucleoid organization in near-native state and revealed an effect of HU surface charges in its overall remodeling during cell growth and environmental adaptation. To define the mechanistic basis of these cellular consequences, we determined the overall organization of HUαα nucleoprotein complexes in solution by small-angle X-ray scattering (SAXS). We find that HUαα organizes DNA differently at different ionic strengths and pHs. By means of macromolecular crystallography (MX), we additionally elucidate HUαα-dependent molecular switches that modulate the DNA organization. Finally, by means of next-generation RNA sequencing (RNA-Seq), we find a link between the nucleoid remodeling and changes in global transcription. This integrative structural study explains how HU can regulate dynamic transformations during nucleoid remodeling as it correlates to global gene regulation.

## Results

**Nucleoid remodeling during *E. coli* growth, stasis, and under stress.** To determine how HU regulates *E. coli* nucleoid architecture during bacterial growth phases and under stress conditions, we imaged wild-type cells (WT) under normal (pH ~7) and acidic (pH ~5) growth conditions as well HUα mutant (HUα^E34K) using SXT. Glu34 was previously identified to be critical to hydrogen bond formation for HUαα–HUαα multimerization that

maintains HUαα/DNA nucleoprotein complexes in vitro[7]. SXT is a high-resolution (60 nm or higher) imaging method that can be applied to fully hydrated, cryo-fixed, and unstained cells in their most native state[16,17]. X-ray absorbance is measured within the "water window" (284–543 eV) where biological materials with differential carbon and nitrogen based composition absorb X-rays an order of magnitude more than the surrounding water. This absorption follows the Beer-Lambert law wherein the photon absorption is linear and a function of the biochemical composition at each spatial position in the cell, generating a unique Linear Absorption coefficient (LAC) measurement for each voxel (3D pixel)[16,17]. Sub-cellular structures are observed non-invasively and segmented based on their LAC differences allowing 3D visualization of their spatial organization (Supplementary Figure 1b). The reproducible and quantitative capabilities of SXT LAC measurements have been previously shown and were successfully utilized in segmenting eukaryotic cells[17].

In contrast to the segmentation of sub-structures in eukaryotic cells, lack of membrane-bound organelles makes it challenging to resolve distinct regions in bacterial cells. Nevertheless, our computer-generated SXT orthoslices (virtual sections) through the WT *E. coli* cells at 60 nm resolution revealed a region of low-LAC with less-bioorganic material (yellow, less condensed) surrounded by a region of high-LAC with more bioorganic material (blue, more condensed) (Fig. 1a). In agreement with conventional fluorescence imaging results, the less-condensed material correlates with the distribution of HU and thus corresponds to the nucleoid[12,18]. The cell periphery enriched in ribosomes[12,19] corresponds to the condensed biomaterial in our SXT imaging and results in LAC values $> 0.4\,\mu m^{-1}$.

The SXT orthoslices of the cells in the lag phase (Fig. 1a) showed a diffused nucleoid region. A histogram of LAC values of the WT cells harvested in the lag phase ($OD_{600}$ 0.2) showed a broad distribution with the maxima at $\sim 0.39\,\mu m^{-1}$ (Fig. 1a). On the other hand, significant separation of the nucleoid region from cell periphery in the exponential growth phase ($OD_{600}$ ~0.5) indicated distinct segmentation of the nucleoid from the cytosol (Fig. 1a). This phase separation was reflected by two maxima in LAC histogram (Fig. 1a, lower panel). WT cells in stationary phase showed more-condensed nucleoid region with a significant shift in the LAC histogram towards higher values with maxima at $\sim 0.41\,\mu m^{-1}$ (Fig. 1a, lower panel) and showed phase separation of nucleoid region. The physical basis of the phase separation emerges by visualization of the 3D reconstruction of the nucleoid region segmented and color-coded based on SXT LAC values. Spatially, the higher-order organization of the nucleoid into distinct domains is akin to the nucleoid macrodomains organization (Fig. 2b, and Supplementary Movie 1)[10,20] where domains appear physically insulated from each other (Fig. 2b and Supplementary Figure 1b). Although, localization of the macrodomains, namely *ori*, *left*, *right*, and *ter* cannot be readily achieved with SXT alone, this first high-resolution near-native visualization of the bacterial nucleoid is consistent with the established global organization of the bacterial chromatin (reviewed in ref. [21]). The 3D reconstructions of the WT cells showed a region with high-bioorganic content located at the center of the cell called here as "nucleoid core" (Fig. 2a, b). The macrodomain architecture of individual chromatin arms varied within a dividing cell and between different cells in similar growth phase (Fig. 2b and Supplementary Figure 1b) indicating dynamic rearrangements of the nucleoid during the cell cycle as suggested previously[12,18,22]. Despite the dissimilarity in the macrodomain arrangement, the nucleoid material in the lag and exponential phases adopt well-segmented interconnected regions separated from the cytosolic material (Fig. 2b). Interconnected regions are easily seen after volume skeletonization (see Methods), whereas distinct domains

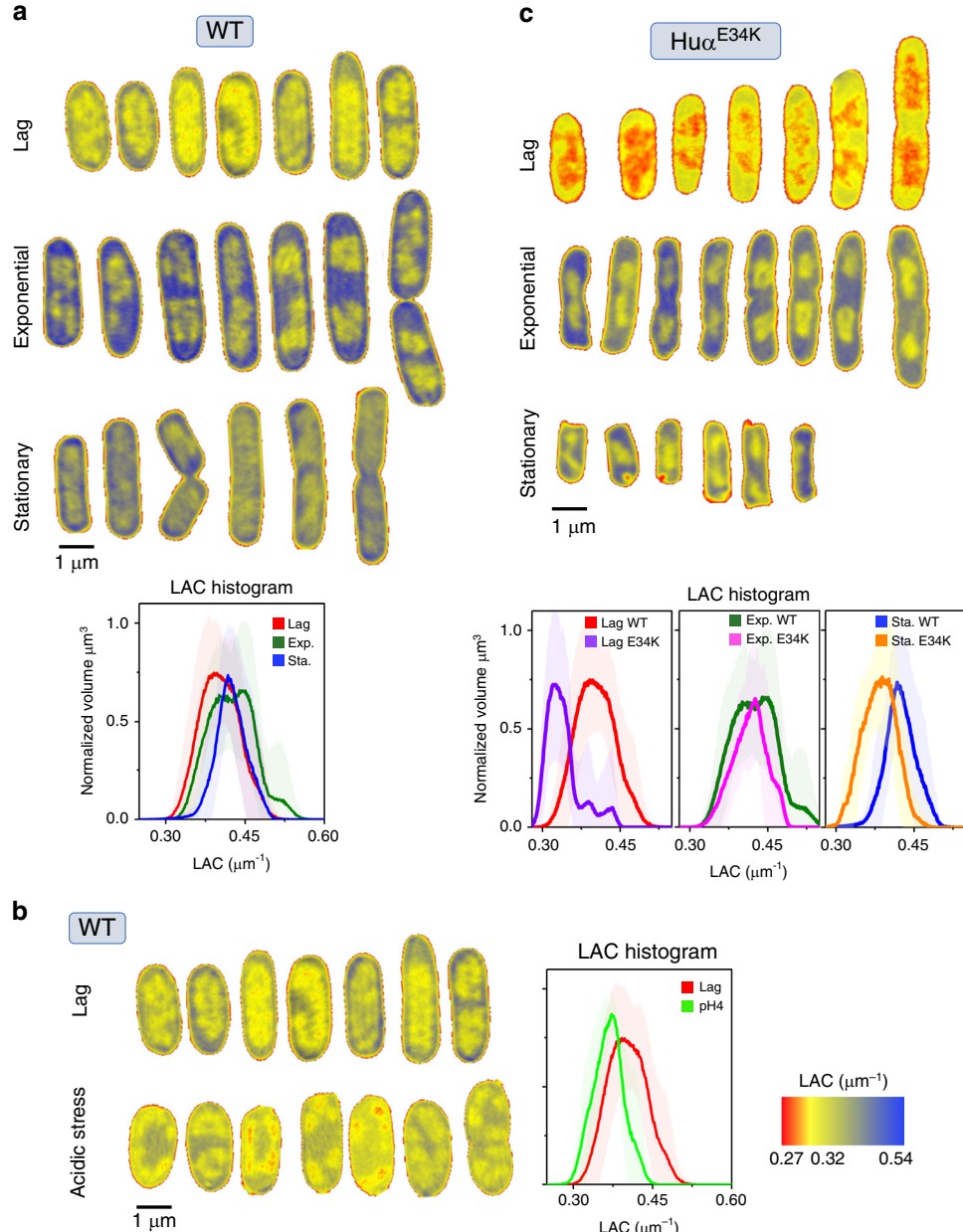

**Fig. 1 Bacterial nucleoid phase transitions under acidic stress and upon HUαα mutation. a–c** Representative orthoslices of *E. coli* wild-type (WT) cells are shown for lag, exponential, stationary growth phases in normal growth media (**a** upper left), under acidic stress (**b** lower left) and upon HUα^E34K mutation (**c** upper right) within the LAC range 0.27–0.54 μm$^{-1}$ (red to blue, shown bottom right inset). Associated LAC histograms present a quantitative measure of nucleoid rearrangement during transition from lag to stationary phase for WT cells in normal growth media (**a** lower panel) and under acidic stress (**b** right panel) and for HUα^E34K mutant cells (**c** lower panel). LAC histograms data are presented as mean values ± SD determined from $n = 10$ independent bacterial cells for each condition.

are absent in the stationary phase (Fig. 2b and Supplementary Movie 1).

Under acidic stress (buffered at ~pH 5) WT *E. coli* cells had significantly longer generation times and reached a final OD$_{600}$ of only ~0.2 similar to lag phase of WT *E. coli* cells grown in normal pH (pH ~7) after 4 hours of growth (Supplementary Figure 1c). Interestingly, the nucleoid region under acidic stress, appeared less-condensed compared with WT *E. coli* cells grown in normal pH as shown by SXT orthoslices and a shift to lower LAC values in the LAC histogram (Fig. 1b). Although the overall morphology of the segmented macrodomains that surrounded the condensed core remained similar to the lag phase WT *E. coli* cells grown in normal pH, the 3D reconstructions showed bulkier cells with

diffused borders between the nucleoid region and the cytosolic material (Fig. 3a).

The pH-dependent nucleoid remodeling prompted us to evaluate the *E. coli* mutant strain—HUα^E34K with mutation of an ionized HUα surface residue critical to HUαα–HUαα multi-merization in vitro[7] for effects on the nucleoid architecture. The mutant HUα^E34K strain had similar growth rate as WT *E. coli* cell at normal pH (Supplementary Figure 1c) while the cell morphology of HUα^E34K cells appeared different from WT *E. coli* cells in exponential and stationary phase (Fig. 1a, c). Orthoslices of HUα^E34K cells in the lag phase showed less-condensed nucleoid region than WT *E. coli* cells grown at normal pH with some recovery of condensation in exponential and

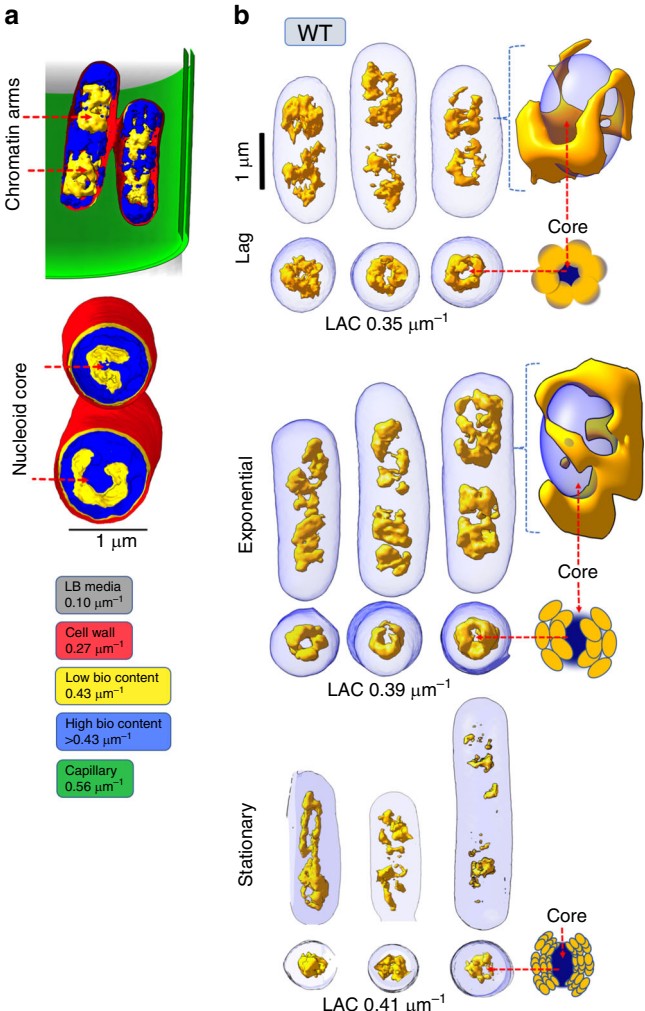

**Fig. 2 Volumetric reconstructions of *E. coli* nucleoid visualized at high spatial resolution.** Soft X-ray tomography volumetric three-dimensonal (3D) reconstructions displays *E. coli* nucleoid architecture and organization at high spatial resolution (~60 nm) (**a**) 3D tomographic reconstructions of *E. coli* cells in a glass capillary are shown for reference. The different regions with distinct LAC are labeled as LB media (gray, LAC 0.10 µm$^{-1}$), cell wall (red, LAC ~0.27 µm$^{-1}$), nucleoid (yellow, ~0.43 µm$^{-1}$), high-bioorganic content (blue, LAC > 0.43 µm$^{-1}$) and capillary (green, LAC 0.56 µm$^{-1}$) (see also Supplementary Figure 1b). **b** Bacterial nucleoid (yellow) is segmented from the cytosol (blue). Representative surface renderings of 3D reconstructions are shown for *E. coli* WT cells for lag, exponential, and stationary phase in normal growth media (see also Supplementary Movie 1). Cartoon shows bacterial nucleoid core (deep blue) surrounded by distinct lobular nucleoid macrodomain architecture (yellow) in normal growth media.

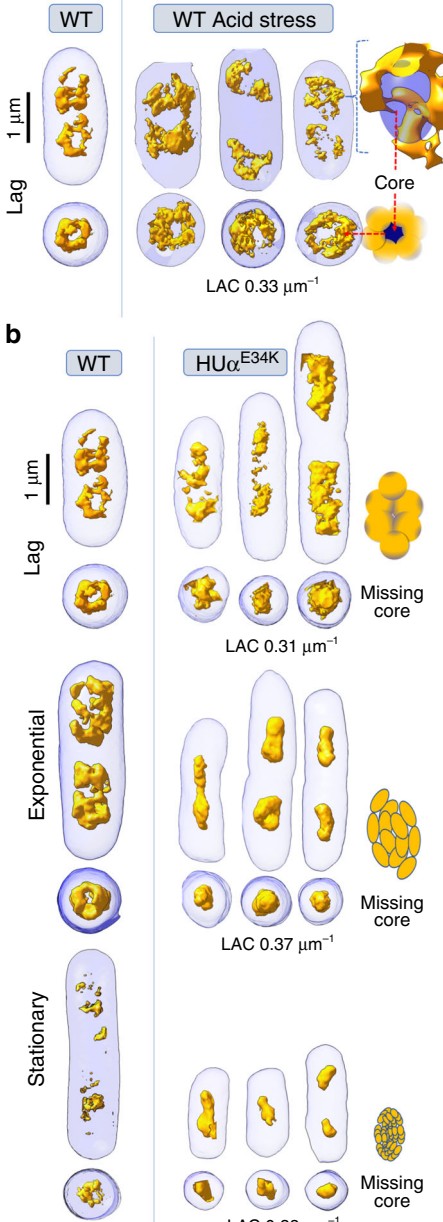

**Fig. 3 Bacterial nucleoid remodeling under acidic stress and upon HUαα mutation.** Bacterial nucleoid (yellow) is segmented from the cytosol (blue). Representative surface renderings of 3D reconstructions are shown comparing **a** *E. coli* WT cells lag phase in normal growth media and under acidic stress and **b** *E. coli* WT cells in normal growth media with HUα$^{E34K}$ mutant cells. Cartoon shows nucleoid core (deep blue) surrounded by distinct lobular nucleoid macrodomain architecture (yellow) in normal growth media that changes to **a** diffused nucleoid region under acidic stress, whereas the lobular bacterial nucleoid has **b** caved-in for the mutant cells.

stationary phase (Fig. 1c). Although, lag phase LAC histogram of HUα$^{E34K}$ cells showed a striking shift to lower LAC values with maxima ~0.32 µm$^{-1}$ similar to WT *E. coli* cells under acidic stress (Fig. 1c, lower panel), 3D reconstructions showed that the nucleoid organization in HUα$^{E34K}$ cells is distinct from WT *E. coli* cells under acidic stress (Fig. 3b).

The distinct phase separation evident as two maxima in the exponential phase LAC histogram of WT *E. coli* cells in normal pH was not seen in HUα$^{E34K}$ cells, whereas there was shift to lower LAC values in the stationary phase LAC histogram (Fig. 1c, lower panel). The remodeling of the nucleoid core in the

HUα$^{E34K}$ cells regardless of the growth phase suggested that HUα is possibly also relevant in maintaining the nucleoid core in addition to the overall nucleoid organization (Fig. 3b). This nucleoid remodeling in HUα$^{E34K}$ cells suggest that the HUα dependent long-range DNA–DNA contacts identified previously[10] were disrupted, presumably as a consequence of direct effect on HUαα multimerization.

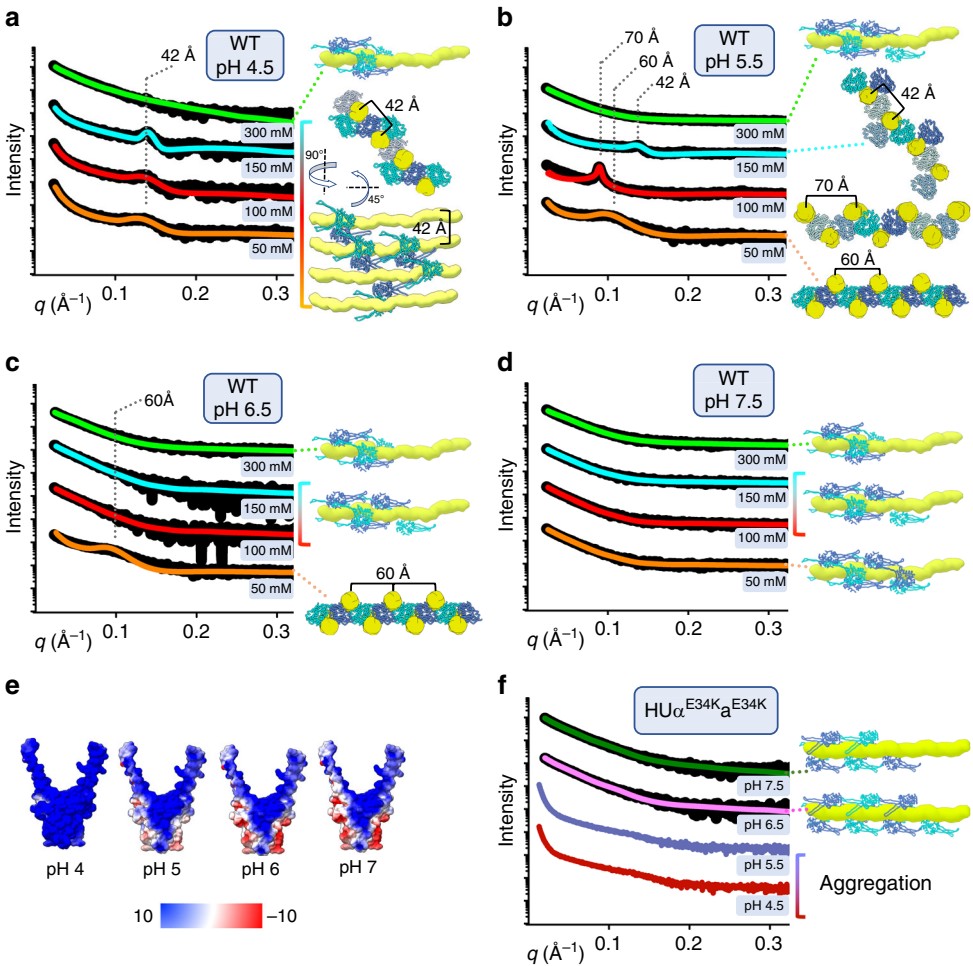

**Fig. 4 HUαα surface charge impact HU–DNA assembly in solution. a–d** Experimental SAXS curves for HUαα −80 bp DNA complex measured at different pH and increasing salt concentration are shown. **a** At pH 4.5, the SAXS match the lamellar structures with DNA spacing 42 Å at 50 mM and 100 mM NaCl that changes to filament-like structures at 300 mM NaCl (see also Supplementary Figure 2a). **b** At pH 5.5, the SAXS curves match the lamellar structures with DNA spacing ~60 Å at 50 mM NaCl, ~70 Å at 100 mM NaCl, and ~42 Å at 150 mM NaCl that changes to filament-like structures at 300 mM NaCl (see also Supplementary Figure 2b). **c** At pH 6.5, the experimental SAXS match the lamellar structures with DNA spacing ~60 Å at 50 mM NaCl that changes to filament-like structures at higher salt concentrations (see also Supplementary Figure 2c, e). **d** At pH 7.5, the SAXS match the lamellar structures with DNA spacing ~60 Å at 50 mM NaCl that changes to filament-like structures at higher salt concentrations (see also Supplementary Figure 2d, f). **e** Electrostatic surface potential for HUαα calculated at the pH = 4.0, 5.0, 6.0, and 7.0. **f** SAXS for HUα^E34K^α^E34K^ −80 bp DNA show aggregation at pH 4.5 and 5.5, and filament-like structures at pH 6.5 and 7.5 (see also Supplementary Figure 3).

**HUαα surface charge regulates DNA bundling**. To understand the mechanism of the HU-dependent cellular outcome of nucleoid remodeling, we characterized the HUαα nucleoprotein complexes in solution by SAXS (Fig. 4). We measured SAXS of HUαα in complex with 80 base pair (bp) DNA of random sequence to understand its overall organization under varied pH and salt concentrations. We deduced that at pH 4.5 and 5.5, HUαα-80bp DNA readily self-assembled into ordered lamellar structures leading to DNA bundling evident by an initial slope of $q^{-2}$[23] and distinct diffraction peaks in the 1D-SAXS profile (Fig. 4a, b and Supplementary Figure 2a, b). We fit the scattering patterns as a random lamellar phase (sheet) with Caille structure factor (see Methods). The presence of the diffraction peak indicated that the physiological salt concentration of 150 mM NaCl supported lamellar assembly at pH 4.5 (Fig. 4a). This assembly with DNA spacing of 42 Å transformed into a filament-like nucleoprotein assembly at 300 mM NaCl concentration (Fig. 4a and Supplementary 2a). We further observed that at pH 5.5, the spacing between the parallel DNAs shifted from 42 Å to 70 Å or 60 Å (Fig. 4b) with a decrease in ionic strength. Further increase

in the pH to 6.5 abolished the formation of lamellar structures with the exception at a sub-physiological salt concentration of 50 mM NaCl (Fig. 4c, d and Supplementary Figure 2c, d). Atomistic models of HUαα-80bp DNA filaments matched the length of ~270 Å defined by the pair distribution function (P(r)) (Supplementary Figure 2e, f) and agreed well with the experimental SAXS data (Fig. 3a and Supplementary Table 2). Our previous mutational analysis[7] together with pH-dependent altering of HUαα–DNA interactions reported here, suggested that HUαα surface electrostatic dictates the hydrogen bond network critical to HUαα–HUαα multimerization that controls DNA bundling (Fig. 4e).

To test this hypothesis, we measured SAXS of HUα surface residue mutant, HUα^E34K^ in complex with 80 base pair (bp) DNA at different pH. We found that the positively charged HUα^E34K^ mutation disrupted DNA bundling in a protein concentration-independent manner and led to the formation of filament-like nucleoprotein structures at pH 6.5, 7.5, and aggregations at pH 5.5, 4.5 with no diffraction peaks (Fig. 4f). The uniform separation of HUα^E34K^α^E34K^ dimers on DNA at low protein

concentration (<2x dimers per 80 bp DNA) appeared as a distinct shoulder in the P(r) function at ~75 Å spacing (Supplementary Figure 3a) and indicates lack of HU-HU multimerization on the DNA. At higher protein concentration (>4× dimers per 80 bp DNA) P(r) shoulder disappears, suggesting HUα$^{E34K}$α$^{E34K}$ covers the DNA completely. Indeed, atomistic models of HUα$^{E34K}$α$^{E34K}$ covering the entire length of the DNA agreed with the SAXS data of the complex with no evidence for bridging or bundling of DNAs (Fig. 4f). Collectively, our SAXS analysis provided support for the critical importance of HUαα surface charge in DNA bundling that manifests into the mesoscale as nucleoid remodeling.

**Molecular switch in HUαα coupling alters DNA bundling**. In order to elucidate the molecular-level mechanism that alters the DNA bundling, we determined crystal structures of HUαα in complex with 19 bp DNA of random sequence at pH ~4.5 and ~5.5 and low ionic strength (~100–150 mM NaCl) (Fig. 5a, b; Supplementary Figure 4a, b and Supplementary Table 1). Structures at both pH showed linear DNA conformation across the HU α-helical "body" (Fig. 5a, b and Supplementary 4a,b) rather than the bent DNA conformation between extended β-ribbon "arms" of HUαα seen with structurally specific DNA interactions[14] (Fig. 5a-inset). Moreover, the non-sequence-specific DNA-binding mode of HUαα in the structures at pH ~4.5 and 5.5 resulted in virtual sliding of the DNA relative to HUαα with out-of-register duplex positions forming distinct HUαα–DNA interfaces. Superimposition of these interfaces revealed the ball-socket joint around the DNA minor grove that permits tilt and twist of HUαα on DNA (Fig. 5c).

We had shown in a previous study that HUαα has two identical faces, either of which interact with DNA through a phosphate lock between the G46-K83 peptide[7]. Combining several asymmetric units (ASU) for either structure at pH ~4.5 and ~5.5, we observed that two parallel DNA strands 42 Å apart are engaged by the phosphate locks around the two faces of an HUαα dimer. The DNA spacing of 42 Å is also observed in solution at low pH and certain ionic strength and is repeated as represented in our DNA bundle model (Fig. 4a). In addition to the interaction of positively charged HUαα body, HUαα arms support intermolecular bundling of DNA. We found that at pH 4.5 the parallel DNA strands were bridged through an additional β-zipper interaction involving the arms of oppositely facing DNA bound HUαα dimers (Fig. 5a). These HUαα dimers held arms through the intermolecular backbone hydrogen bonds between the highly conserved residues R61 and N62 (Supplementary Figs. 1a and 5a, c). The flexible elbow joint (residues A57, A73) positioned the arms to form the β-zipper reinforcing the DNA bundle (Supplementary Figure 5c). In contrast, HUαα arms did not supplement the DNA bundling at pH 5.5 (Fig. 5b). Interestingly, another major difference between the structures at different pH was in the coupling of two HUαα dimers connecting additional parallel DNA strands. In the structure at pH ~4.5, two HUαα dimers couple around the minor groove of a DNA strand through intermolecular hydrogen bonds between residues E34K and K37 (Fig. 5a and Supplementary Figure 5a). In contrast, the intermolecular hydrogen bonds were disrupted in the structure at pH ~5.5 uncoupling the HUαα dimers by shifting one dimer ~3 Å away (Fig. 5d, left panel and Supplementary Movie 2). Instead, we observed alternative coupling of HUαα dimers through intermolecular hydrogen bonds between residues D8 and Q5 (Fig. 5b and Supplementary Figure 5b). The alternative coupling and lack of arm bridging transitioned the DNA–DNA spacing in the crystal to 70 Å (Fig. 5b) that corroborated our observations in solution at similar pH and higher ionic strength

(Fig. 4b). We concluded that the altered surface charge owing to change in protonation state of residue E34 (Fig. 4e) altered intramolecular hydrogen bond that leads to the formation of alternative HUαα–HUαα coupling through residues D8 and Q5 at pH 5.5 and explained the shift in nucleoprotein organization in solution. The crystal structure of surface residue mutant HUα$^{E34K}$α$^{E34K}$ with the 19 bp DNA further validated the importance of HUαα–HUαα coupling in DNA bundling. The lack of hydrogen bond network caused a separation between the HUα$^{E34K}$α$^{E34K}$ dimers to even larger distances of ~5 Å. (Fig. 5d, right panel and Supplementary Movie 2). No alternative HUα$^{E34K}$α$^{E34K}$- HUα$^{E34K}$α$^{E34K}$ coupling similar to that for HUαα/19 bp DNA at pH ~5.5 was observed either which correlated with the observation of filament structures of HUα$^{E34K}$α$^{E34K}$-80 bp DNA in solution (Fig. 4f and Supplementary Figure 3a). The phosphate lock-based HUα$^{E34K}$α$^{E34K}$-DNA interaction interfaces were retained and dictated the tilt and twist around the DNA (Supplementary Figure 5d).

In summary, combing crystal structures and solution states we showed that HUαα-body and HUαα-arms based multimerization is dependent on pH and salt concentrations. Furthermore, we revealed a role of HUαα surface charge in HUαα-HUαα coupling-dependent adaptability of HU–DNA interfaces that dictates the transition between DNA bundles and filaments.

**HU-dependent nucleoid remodeling correlates to global gene expression changes**. To show that nucleoid remodeling, we observed under different growth phases and acidic stress condition, is related to changes in global gene expression, we measured gene expression in using RNA-seq (see Methods). SXT imaging showed that nucleoid material is less condensed in the lag phase compared with the log phase. By differential expression analysis of genes between the log phase and lag phase in WT cells, we found that 199 genes were upregulated, whereas 73 genes were downregulated in the lag phase (Fig. 6a and Supplementary Data 1). We interpret that the less-condensed nucleoid in the lag phase correlates with higher gene expression and the more-condensed nucleoid in the growth phase strongly correlates with the lower expression of the same genes. This link between nucleoid remodeling and gene expression in combination with specific gene regulatory mechanisms may be critical for the global gene expression reprogramming to prepare E. coli for the transition into the exponential growth[24,25]. SXT imaging also showed that the nucleoid region was less condensed in WT cells grown under acidic conditions compared to those grown at pH 7.0. In contrast to a significant separation of nucleoid region from the cytosolic material at pH 7.0, we observed diffused borders between the nucleoid region and the cytosolic material at pH 5.0. By differential gene expression analysis in WT cells between the normal and acidic growth conditions, we found that 245 genes were upregulated and 161 genes were downregulated in cells grown under the acidic condition (Fig. 6b and Supplementary Data 2). In agreement with previous studies[26], we observed upregulation of genes, such as asr in our analysis, are important for survival under acid stress. Although differential expression of many of the genes could result from a direct response to acidic stress through specific TFs-mediated gene regulation, we found that only 175 genes were upregulated at pH 5.0 when HU was absent, suggesting that differential expression under acidic condition is at least partially dependent on HU (Fig. 6c and Supplementary Data 3).

We have demonstrated that HUαα–HUαα coupling mediated by the surface charge residues is critical for the higher-order organization of DNA into bundles and changes in E34 protonation state owing to a change in pH or the E34K mutation

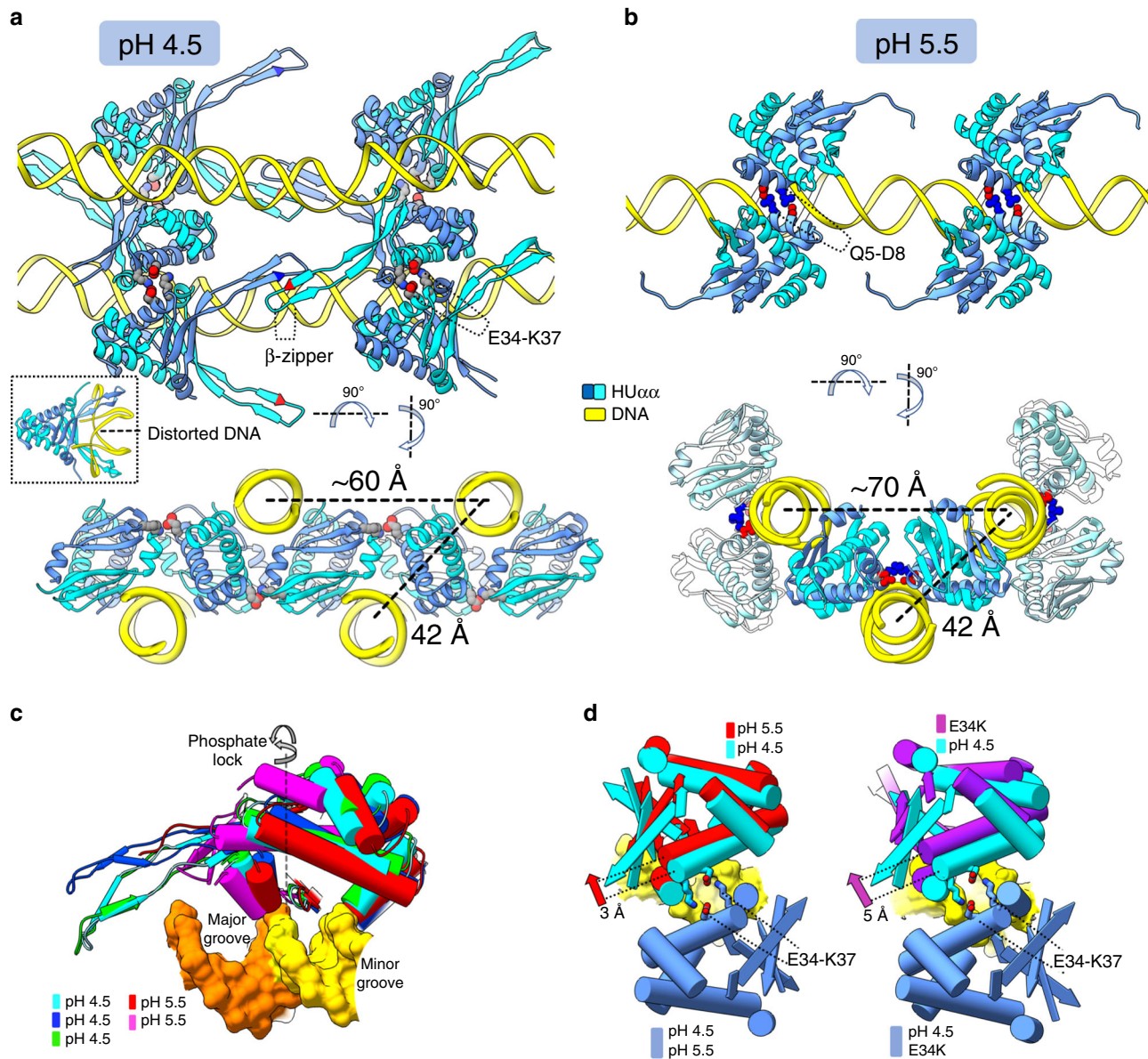

**Fig. 5 Structural details of HUαα-DNA assembly reveal promiscuity in HUαα-DNA and HUαα-HUαα interfaces. a–b** Orthogonal views of HUαα-DNA structures at pH 4.5 and pH 5.5 combining several asymmetric units (see also Supplementary Figs. 4 and 5a–c). **a** At pH 4.5, two distinct HUαα-HUαα interaction interfaces are seen–the highly electrostatic E34-K37 interface and the HUαα-HUαα arm β-zipper (some DNA strands have been removed for clarity). Inset–previously reported crystal structure of HUαα with artificially distorted DNA (PDBID: 1P71) with HUαα-arms intercalating between base pairs in the DNA minor grooves. **b** At pH 5.5, HUαα-HUαα interaction interface between residues Q5 and D5 is highlighted (some DNA strands have been removed for clarity). Distances between two sets of parallel DNA that correspond to diffraction peaks observed in solution scattering are highlighted. **c** Non-specific DNA-binding mode of HUαα results in several non-superimposable HUαα-DNA interfaces relative to the phosphate lock. Superimposition of the distinct HUαα-DNA interfaces at pH 4.5 and pH 5.5 reveals the ball-socket joint around the DNA minor grove. **d** Superimposition of the HUαα-HUαα interface from the crystal structure of HUαα-DNA at the pH 4.5 with interface seen in the crystal structure at the pH 5.5 (left panel) or crystal structure of HUα$^{E34K}$α$^{E34K}$-DNA (right panel). Unpaired HUαα dimers at pH 5.5 and HUα$^{E34K}$α$^{E34K}$ dimers shows the 3 Å and 5 Å shift along the DNA minor grove relatively to the coupled HUαα dimers at pH 4.5 (see also Supplementary Figure 5d and Supplementary Movie 2).

could result in the reorganization of DNA from bundles into filaments. We hypothesized that a change in the HUαα composition under different growth phases, which is known to occur in *E. coli*, and or the changes in E34 protonation state owing to changes in cytosolic pH could account for gene expression changes we observed. To test our hypothesis, we measured the gene expression of HUα mutant (HUα$^{E34K}$) cells with or without the presence of HUβ under normal and acidic growth conditions. Although *E. coli* uses an active mechanism of

pH homeostasis[27,28], we expected that external acidic conditions would cause a rapid decrease in cytoplasmic pH[29]. Surprisingly, the E34K mutation in HUα did not influence the gene expression both in the presence or the absence of HUβ under both normal growth conditions and acidic stress conditions (Fig. 6d–f and Supplementary Data 4–5). We speculate that the higher-order organization through HUαα–HUαα coupling mediated by the residue E34 plays an architectural role in the nucleoid structure without affecting gene expression. We propose that HU-mediated

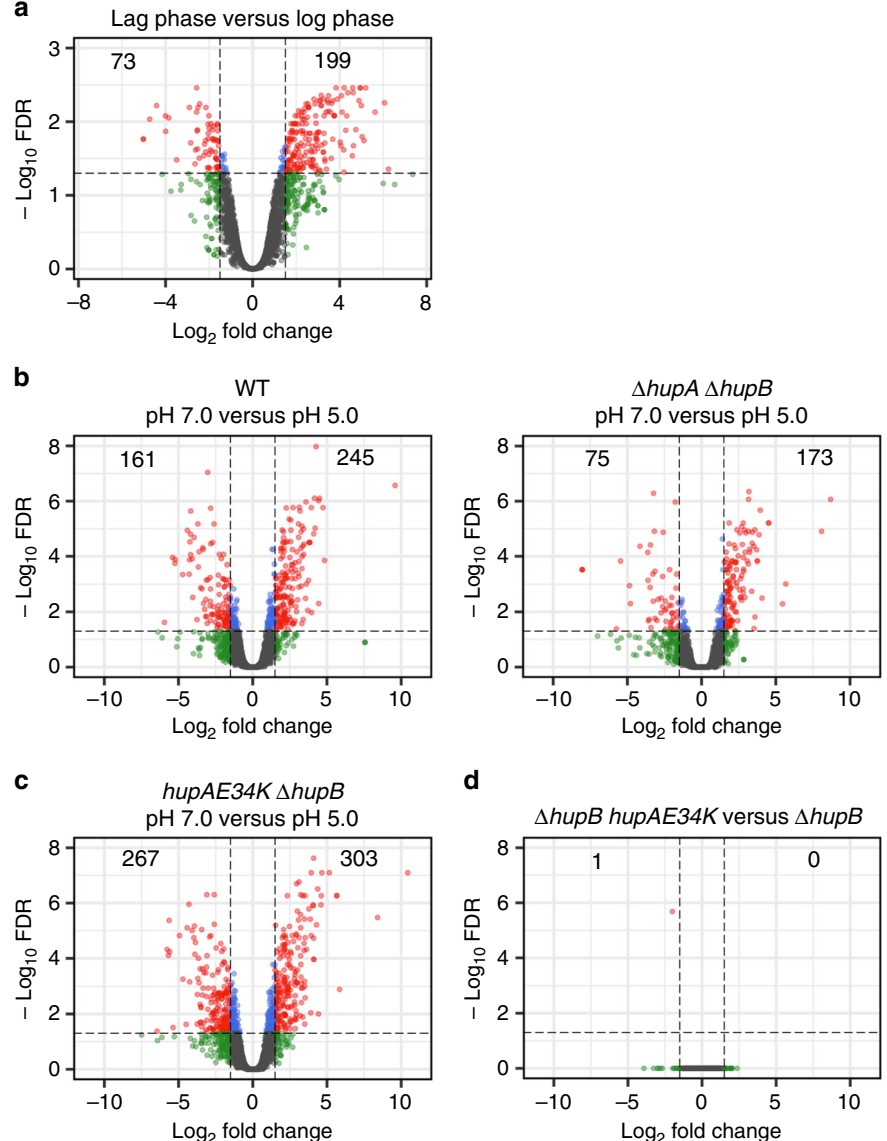

**Fig. 6 Global gene expression owing to lag phase, acidic growth, or HU mutations.** Volcano plots showing log2-fold change (logFC) in transcript abundance plotted against $\log_{10}$ FDR (false discovery rate). Each dot represents a gene. Vertical bars indicate logFC cutoff of −1.5 and 1.5 and a horizontal bar indicates FDR cutoff of $-\log_{10} 0.05$. Orange dots indicate differentially expressed (DE) genes with absolute logFC values >1.5 and FDR values <0.05. **a** Differential gene expression between log phase and lag phase in WT strain (SCV96). A positive logFC value indicates higher expression of a gene in lag phase relative to log phase, whereas a negative logFC value indicates lower expression of a gene in lag phase (see also Supplementary Data 1). **b–e** Differential gene expression between pH 7.0 and pH 5.0 in ΔhupAΔhupB (SCV27), hupAE34K hupB + (SCV56), and hupAE34KΔhupB (SCV85). A positive logFC value indicates higher expression of a gene at pH 5.0 relative to pH 7.0, whereas a negative logFC indicates lower expression of a gene at pH 5.0. Cells were harvested in lag phase at both pH (see also Supplementary Data 2–4). **f** Differential gene expression between the strain harboring wild-type HUβ but lacking HUβ (SCV19) and strain harboring HUαE34K but lacking HUβ (SCV85). A positive logFC indicates higher expression of a gene in SCV19 relative to SCV85, whereas a negative logFC indicates higher expression of a gene in SCV85 (see also Supplementary Data 5).

restraining of supercoils may play a direct role in controlling global gene expression.

## Discussion

3D genome organization is crucial for cellular behaviors and genetic transmission. 3D genome is also highly dynamic[30,31] and its reconfiguration through an influence on gene expression is important, at least in some eukaryotes, for cellular differentiation during embryonic development[32]. In eukaryotic cells, gene expression is linked to an unwinding of DNA, and the loose euchromatin marks the regions of global gene expression[17,33].

In the prokaryotic world, the nucleoid is a discrete, internally organized unit wherein the overall arrangement of DNA regulates gene expression globally[34].The *E. coli* nucleoid, the functionally and spatially organized form of a single 4.6 megabase-pairs size haploid chromosome, serves a simple model to study 3D configuration of a genome and its dynamic reconfiguration during environmental changes. However, high-resolution visualization of the nucleoid to better understand organizational features as well as unraveling molecular mechanisms, especially involving biochemical factors such as NAPs, responsible for maintaining 3D genome architecture has presented major challenges. Using the X-ray-based imaging technology—SXT, we determined the

architecture of the bacterial nucleoid region in near-native state at 60 nm resolution. We observed that the nucleoid region consisted of a condensed core surrounded by spatially isolated domains or macrodomains consistent with previously proposed features of the nucleoid. Our studies validated the role of HUαα, the most-abundant NAP in promoting megabase range DNA communication for maintaining the architecture of the nucleoid macrodomains as was previously identified by Chromosome Conformation Capture Analysis (3 C)[6,10]. We could also visualize dynamic reorganization of the domain in response to environmental changes and by mutation of HUα, which is in conjunction with other factors, like DNA-specific NAPs, RNA polymerase modulators, and topoisomerases such as DNA gyrase[35–37]. Although, we observed distinct sub-regions such as nucleoid core and macrodomains, within the nucleoid region in E. coli, their exact composition and protein:nucleic acid ratio cannot be determined with SXT alone. Nevertheless, our observations correlate with previous observations of longitudinal density bundling and radial confinement of E. coli nucleoid[18,38,39].

Similar to histones in eukaryotes, HU that binds non-sequence-specific DNA, is essential for bacterial nucleoid condensation that relates to global gene expression[3,40]. Our gene expression analysis indicates that the nucleoid remodeling may be crucial for the reprogramming of gene expression for environmental adaptation and provide a testable model of the relationship between HU structural assemblies, nucleoid architecture, and gene expression.

Our characterization of the hierarchical organization of HUαα–DNA nucleoprotein complexes in solution utilizing SAXS provided evidence for DNA bundling similar to previous reports[7,41]. We further found that the DNA bundle transformed to rigid filament-like structures by varying pH and ionic strength similar to the HU–DNA nucleoprotein complexes formed at high HUαα:DNA ratios (>30 HUαα per 100 bp DNA)[42,43]. Considering uniformly distributed HU across the nucleoid as previously shown[11] with only one HU dimer per ~150 bp DNA, we propose that sparsely bound HUαα can bundle multiple DNA segments in the nucleoid while still allowing efficient exchange of other DNA supercoil modifiers. Furthermore, we suggest that the transition from a DNA bundle to a filament structure depending on HU/DNA ratio, environmental pH, or ionic strength contributes to the nucleoid remodeling.

In the past, crystal structures of several HU–DNA complexes have been studied. HU–DNA complexes with structurally aberrant DNA proposed DNA bending involving HU arms as a mode of DNA binding[14]. In addition, multiple other models proposed HU–DNA complex involving single DNA-duplex strand in a nucleoprotein complex[44–47]. In contrast, in our crystal structures we observed that a two linear DNAs are bound across two opposite faces of HUαα dimer. In addition, we found that multiple HUαα dimers can bundle DNA strands through HUαα–HUαα coupling involving HU body reinforced by HU arms. We further revealed that changes in HU surface charge by varying pH, salt, or surface charged mutation[7] cause repulsive separation of HU dimers along the DNA-transforming DNA bundles into filaments. Altering of the DNA bundling explains nucleoid remodeling in rapidly growing E. coli cells and further evidence the role of HUαα in previously identified long-range DNA interactions[10].

Although intercalation of the HUαα arms with structurally specific aberrant DNA may be a lock-and-key mechanism for binding damaged DNA sites[14], the dynamic HUαα–HUαα coupling is a universal mechanism for the binding sequence-independent DNA. HUαα–HUαα coupling that involves both, HUαα body and arms, distinguish general DNA condensation complexes from specific aberrant DNA complexes. Collectively, our results defined a molecular mechanism whereby changes in HUαα–HUαα multimerization alter DNA bundling affecting

nucleoid remodeling that can change global gene expression in E. coli to synchronize the genetic response to external conditions. Our multi-scale study supports the previously presented idea that for HU-directed bacterial nucleoid remodeling its molecular functionality is key to its macroscopic behavior and manifests at the nanoscale extending into the mesoscale.

Future studies using correlative light and SXT microscopy to label specific sites in the nucleoid will provide more insights into nucleoid dynamics that regulates gene expression reprogramming during environmental adaptations. Nevertheless, this study has broad significance in expanding the understanding of the nucleoid architecture in bacteria and bring insights into HU-mediated nucleoid condensation, leading to concerted changes in the basal transcription program. These observations make HU interactions an attractive target for controlling not only pathogenesis but also, microbial systems in general.

## Methods

**Construction of bacterial strains.** HUα[E34K] mutant strain was generated by Lambda red recombineering[48] in two steps using the plasmid pSIM6[49]. In the first step, wild-type hupA open reading frame (ORF) was replaced with a pBAD-ccdB-kan cassette that expresses CcdB toxin under the arabinose inducible promoter pBAD and encodes a kanamycin resistance protein. The recombinants were selected on LB agar plates containing 1% glucose and 30 μg ml⁻¹ kanamycin. The recombinants were verified by PCR using the primers that bind outside the hupA gene as well as by the inability of the recombinants to grow normally on LB agar plates supplemented with 0.02% arabinose that induces the toxin CcdB. In the second step, a synthetic double-stranded DNA (synthesized by Integrated DNA Technologies–https://www.idtdna.com/pages) encoding hupAE34K with 46 nucleotides of homology to the 5' and 3' end of the hupA gene was recombined at the hupA locus. The recombinants were selected on LB agar with 0.02% arabinose and subsequently were verified by Sanger sequencing. After sequence verification, the plasmid pSIM6 was removed by repeatedly growing the strain at 37 °C and checking for ampicillin sensitivity. The final strain SCV56 with mutation hupAE34K was used for further RNA-seq experiments.

For RNA-seq, the following additional strains were constructed. The ΔhupA (SCV18) and ΔhupB (SCV19) strains were made by Lambda red recombineering using the plasmid pSIM6 in the two steps. The selection of the recombinants was done using the steps described for SCV56. In the first step, the hupA ORF from codons 2 to 60 or the hupB ORF from codons 2 to 64 was replaced with the pBAD-ccdB-kan cassette. In the second step, a 70 nucleotides long single-stranded oligonucleotide corresponding to sense strand was recombined. The sequence of the oligonucleotide for hupA is -

GCAATTTTGATTTCTTTACCGGTCTGCGGGTTGCGCATAAGTTATCCT TACAATGTG TTTATCGCTTGCT

for hupB is -

ACTTTAGCAGCAGCGATGGTGATCTCTTTACCGGTCACTCTTCTCTTC CTCTTTATAATTTATATCGCAC.

The strain lacking both HU subunits hupA and hupB was constructed by transducing the hupB11 allele[50] into the strain SCV18 by standard P1vir transduction. The strain hupAE34K ΔhupB::CamR (SCV85) was constructed by transducing the hupB11 allele into SCV56. The hupA deletion of the strain SCV18 was repaired and restored to wild-type hupA gene in two steps to create the strain SCV96 that served as a parent wild-type strain. In the first step, the pBAD-ccdB-kan cassette was introduced at the hupA locus and a wild-type copy of hupA gene amplified from MG1655 was recombined in the second step. The wild-type status of hupA gene was verified by Sanger sequencing.

**HUα and HUαE34K protein expression and purification.** HUα and HUα mutant (HUα[E34K]) genes cloned in expression vector pET15b (Novagen) with tobacco etch virus (TEV) protease-cleavable site following the His6 site were purchased from Genewiz. The plasmids were transformed into BL21(DE3)/pLysS strains. Overnight preculture were grown in Luria Broth (LB) media with 100 μg mL⁻¹ ampicillin and 25 μg mL⁻¹ chloramphenicol antibiotics at 37 °C. Next day, cells from overnight culture were grown in 2 L fresh LB media at 37 °C, up to an optical density of ~0.60–0.8 followed by induction with 0.5 mM isopropyl 1-thio-D-galactopyranoside. The cells were harvested after 4 h by centrifugation and resuspended in buffer A (10 mM HEPES pH 7.9, 0.5 M NaCl, 5 mM imidazole) with an additional 0.1% Triton X-100, 2 mM PMSF, 0.5 mg mL⁻¹ lysozyme, 50 μg mL⁻¹ DNase I, 20 μM CaCl₂, and 4 mM MgCl₂. Cells were lysed by passage through a constant cell disruptor (Constant Systems Ltd.) at 5 °C and 21 kpsi. After lysis, cellular debris was pelleted by centrifugation at 20,000 × g. Cleared cell lysate was batch-bound onto Ni-NTA resin (Qiagen) by rocking overnight at 4 °C. Resin was batch washed in buffer A, then washed with 10× CV buffer A + 20 mM imidazole. Elution was carried out with 10× CV buffer A + 250 mM imidazole. The Ni-NTA eluate was dialyzed overnight in buffer A. Next, the His tag was removed using TEV

protease[51]. The cleaved protein was further purified by passing on Ni-NTA column to remove the His tag as well as TEV protease that stayed bound to the column. The flow through was dialyzed into buffer B (2 mM HEPES pH 7.9, 20 mM NaCl) in 3.5 K MWCO Slide-A-Lyzers (Thermo Scientific). Dialysate was then purified by cation exchange chromatography on a HiTrap SP HP column (GE Healthcare). Protein was bound onto the column then eluted with a gradient starting from buffer B to buffer B + 1 M NaCl. Purified protein at concentration of ~1–2 mg ml$^{-1}$ was flash frozen and stored at −80 °C. Protein concentration of all purified HU was determined by the Bradford and Lowry Assay. Bradford assay: 1.0 mL of Bradford reagent (Bio-Rad) was mixed with 20 μL protein and allowed to react for 10 min at room temperature. Absorbance at 595 nm was then measured on an Agilent 8453 UV-Visible spectrophotometer. Bovine serum albumin (Thermo Scientific) was used as a standard. Concentrations of HU were verified with a modified Lowry assay (Thermo Scientific).

**Soft X-ray tomography**. *E. coli* strain MG1655 with wild-type HUα and HUα$^{E34K}$ mutant strain (SCV56) were used for SXT experiments. For SXT experiments, strains were grown in 5 ml LB media at 37 °C. Cells were put on ice and promptly cryo-immobilized after reaching OD$_{600}$ 0.2 for lag phase, 0.5 for exponential phase and >1.0 for stationary phase. Wild-type *E. coli* cells (MG1655) were also grown in 5 ml LB media at 37 °C buffered to pH ~5 (buffered with 0.1 M sodium acetate) and were put on ice at time intervals at which normal growth media reached lag, log, and stationary phase. Cells grown at low pH (~5) only reached an OD$_{600}$ of ~0.2–0.3 corresponding to lag, log, and stationary phases of normal growth rate.

Cells were transferred to thin-walled glass capillary tubes and vitrified by plunge-freezing in ~90 K liquid propane prior to being mounted in the cryogenic specimen rotation stage of the XM-2 Soft X-ray microscope at the National Center for X-ray Tomography located at the Advanced Light Source (LBL, Berkeley CA)[52]. The microscope is equipped with Fresnel zone plate-based objective lens (micro zone plate (MZP)) with a spatial resolution of 60 nm. Imaging was performed with the specimens in an atmosphere of helium gas cooled by liquid nitrogen. For each data set, 90 projection images were collected sequentially around a rotation axis in 2° increments, giving a total rotation of 180° with exposure time varied between 150 and 300 ms (average 200 ms), depending on the capillary thickness. Images were collected using a Peltier cooled, back-thinned and direct illuminated 2048 × 2048 pixel soft x-ray CCD camera (Roper Scientific iKon-L, Trenton, NJ, USA). Projection images were normalized, aligned and tomographic reconstructions calculated using iterative reconstruction methods in AREC-3D package. The 3D X-ray tomograms were manually segmented in CHIMERA[53], and used to reconstruct volumes and measure voxel values (i.e., absorption values in volume element of the reconstructed data) to calculate linear absorption coefficients (LAC). LAC histograms of 10 cells for each condition were determined using AMIRA (FEI Visualization Sciences Group) and mean and standard deviation of LAC histogram for each condition (control condition, low pH and HUαα mutant strains) were calculated in ORIGIN (https://www.originlab.com/). SXT orthoslices (Fig. 1) and volumetric reconstructions (Figs. 2 and 3) of nucleoid region were displayed in CHIMERA[53]. Skeletonize map of nucleoid region (Fig. 2) was calculated and displayed in CHIMERA[53] by using Gaussian filtering.

**Transcription profiling**. For RNA-Seq, strains were first grown overnight in LB (Lennox) broth and then overnight cultures were diluted 1:1000 in fresh regular LB broth (pH 7.0) or LB with pH 5.0. LB with pH 5.0 was prepared as follows: For 200 ml, 35 ml of 0.2 M citric acid, and 65 ml of 0.2 M sodium citrate were mixed, the components of LB (Lennox broth) were added, and then the volume was doubled with Milli Q water. The final pH of the media was pH 5.0. Cells were harvested at OD600 0.2–0.3, which we refer to as lag phase and 0.4–0.5, which we refer to as log phase in this study. For total RNA isolation, a frozen pellet of cells was resuspended in 1 ml TRIzol reagent (Life Technologies) to homogenization and incubated at room temperature for 5 min. To the resuspension, 0.2 ml chloroform was added and mixed by inverting the tube for 15 seconds. The mixture was incubated at room temperature for 10 min and then centrifuged at 20,000 × g for 10 min at 4 °C. After centrifugation, ~0.6 ml of the upper phase was transferred to a new centrifuge tube containing 0.5 ml isopropanol. The mixture was incubated at room temperature for 10 min and then centrifuged at 20,000 × g for 15 min at 4 °C. After centrifugation, the supernatant was discarded and the pellet was washed twice with 1 ml 75% ethanol, made with Diethyl pyrocarbonate (DEPC)-treated water, by centrifugation at 13,544 × g, for 5 min at 4 °C. After the second wash, the tube was left open for 10–15 min at room temperature to dry the pellet. To the pellet, 50 μl DEPC treated water was added and the tube was left at 37 °C for 10–15 min and then the pellet was fully resuspended using a pipette. DNA was removed using TURBO DNA-free Kit (Invitrogen). Quality of total RNA was determined by electrophoresis on the TapeStation system (Agilent). Paired-end sequencing libraries were prepared with 2.5 μg of total RNA using Illumina TruSeq Stranded Total RNA library prep workflow with Ribo-Zero. Samples were pooled and sequenced on HiSeq4000. Samples were barcode demultiplexed allowing one mismatch using Bcl2fastq v2.17.

The samples had 18–24 million pass filter reads with >89% of bases above the quality score of Q30. The reads were trimmed for adapters and low-quality bases using Cutadapt software[54]. Alignment of the reads to the *E. coli* K12 MG1655 reference genome and the annotated transcriptome was done using STAR[55].

Transcript abundances were calculated by RSEM[56], and differential expression analysis was done using the glmTreat function in edgeR[57]. We identified differential expression based on false discovery rate (FDR) cut off of 0.05 and log2 (log with base 2) fold change of 1.5. The RNA-Seq data have been deposited to the Gene Expression Omnibus (GEO) data base and can be accessed with the GEO accession number: GSE134667.

**DNA purification**. DNA's for crystallographic and solution scattering studies were purchased from Integrated DNA Technologies (https://www.idtdna.com/pages). 19 bp DNA (5′-TTCAATTGTTGTTAACTTG-3′), 19 bp DNA with 1 bp overhang (5′-CGGTTCAATTGGCACGCGC-3′) and 80 bp DNA (5′-AATGAGGTAACAA CGAAAGCAGATGATAGCTGCTTATCAATTTGTTGCAAACAAGTAGCCGC GCCCAATGAGGTAACAAT-3′) were dissolved and annealed in water. Annealed DNAs were purified on a size exclusion chromatography on a Superdex75 column in 20 mM Tris-HCl pH 7.0, 50 mM NaCl.

**Crystallization**. Crystals of HUαα or HUα$^{E34K}$α$^{E34K}$ in complex with native DNAs were grown by hanging-drop vapor diffusion. Various lengths of the native DNA with and without 1 bp overhangs were tested. In all, 19 bp DNA with blunt end and 19 bp DNA with 1 bp overhang on both side was selected for crystallization experiments with HUαα or HUα$^{E34K}$α$^{E34K}$. HUαα was dialyzed into 10 mM Bis-Tris pH 4.5, 150 mM NaCl or into 10 mM Bis-Tris pH 5.5, 100 mM for crystallizing at low pH 4.5 or higher pH 5.5, respectively and concentrated to ~17 mg ml$^{-1}$. The 19 bp DNA was buffer exchanged into 20 mM Tris pH 7.0, 50 mM NaCl and was concentrated to 10 mg ml$^{-1}$. Well diffracting crystals were obtained with 0.6 μl 0.1 M Na-malonate, pH 4.0 12% PEG 3350, 0.6 μl of 19 bp DNA, and 0.6 μl of HUαα dehydrated over 0.1 M Na-malonate, pH 4.0 12% PEG 3350 (crystals at pH 4.5) and with 0.02 M ZnCl$_2$, 20% PEG 3350 (measure pH 5), 0.6 μl of 19 bp DNA, and 0.6 μl of HUαα dehydrated over 0.02 M ZnCl$_2$, 20% PEG 3350 (crystals at pH ~5.5). HUα$^{E34K}$α$^{E34K}$-DNA was dialyzed into 10 mM Bis-Tris pH 6.5, 50 mM NaCl and concentrated to ~17 mg ml$^{-1}$. The 19 bp DNA with 1 bp overhang was buffer exchanged into 20 mM Tris pH 7.0, 50 mM NaCl and was concentrated to 10 mg ml$^{-1}$. Well diffracting crystals were obtained with 0.6 μl 0.1 M Na-malonate, pH 5.0, 12% PEG 3350, 0.6 μl of DNA, and 0.6 μl of HUα$^{E34K}$α$^{E34K}$ dehydrated over Na-malonate, pH 5.0, 12% PEG 3350. The best diffraction crystals grew in the drops produced by streaking protein into the DNA drop. Crystals grew in 2–3 days at 20 °C.

**Data collection and structural analysis**. Structures of HUαα-DNA and HUα$^{E34K}$α$^{E34K}$-DNA complexes were solved by molecular replacement with X-ray diffraction data collected from cryo-cooled crystals at the SIBYLS Beamline BL 12.3.1[58] and BL 8.3.1 of the Advanced Light Source (ALS, Berkeley, CA). The data were indexed, integrated and scaled with XDS[59] and the phases were defined by molecular replacement in PHASER[60] using the refined structure of HUαα (PDBID: 4YEX)[7]. Mathews coefficient was utilized to determine the contents of the ASU[61]. The molecular replacement phases yielded clear density for double-stranded DNA. Ideal DNA helices were placed manually into the density and the model was refined using Phenix Refine[62]. The side chains were built using sigmaA weighted 2Fo-Fc and Fo-Fc maps in COOT[63] (Supplementary Table 1). Structures were visualized in CHIMERA[53]. Electrostatic surface potential was calculated by Poisson-Boltzmann electrostatic calculations at the pH = 4.0, 5.0, 6.0, and 7.0[64].

**SAXS data collection and evaluation**. SAXS data were collected at the ALS beamline 12.3.1 LBNL Berkeley, California[58] at wavelength 1.03 Å using a Pilatus 2M detector with the sample-to-detector distance of 1.5 m, resulting in scattering vectors, $q$ ranging from 0.01 Å$^{-1}$ to 0.35 Å$^{-1}$. The scattering vector is defined as $q = 4\pi \sin\theta/\lambda$, where $2\theta$ is the scattering angle. All experiments were performed at 10 °C. For protein–DNA assemblies, protein HUα$^{E34K}$α$^{E34K}$ was mixed with 80 bp DNA ~5 min prior data collection at the HUα$^{E34K}$α$^{E34K}$/DNA ratios as indicated in Supplementary Figure 3. SAXS data were collected from 30-μl samples of HUαα, HUαα-80bp DNA, HUα$^{E34K}$α$^{E34K}$, and HUα$^{E34K}$α$^{E34K}$-80 bp DNA with 3 s X-ray exposures for 5 mins continuously exchanged into different buffer conditions (please see SASBDB data base for the buffer composition). Integral of ratio of the background of frames collected provided a signal plot for accurate buffer subtraction. SAXS curves were merged and analyzed using the program SCÅTTER. The SAXS profiles of HUαα-80 bp DNA under certain buffer conditions showed diffraction peaks defined by d-spacing between parallel DNAs, indicating formation of a crystalline phase in solution. These SAXS profiles cannot be used to obtain the radius of gyration $R_g$ or the $P(r)$ function. The SAXS profiles for HUαα, HUαα-80bp DNA, HUα$^{E34K}$α$^{E34K}$, and HUα$^{E34K}$α$^{E34K}$-80bp DNA complexes collected presently, which did not show diffraction peaks were investigated by comparing radius of gyrations $R_g$ derived by the Guinier approximation[65]

$$\ln I(q) = \ln I(0) - \frac{1}{3}q^2 R_g^2 \qquad (1)$$

with the limits $q*R_g < 1.3$. Aggregation-free states of samples were investigated by defining linear region in the Guinier plots (Supplementary Figs. 2e, f and 3b). Cross-sectional radius of gyration ($R_c$) and Porod volume[23] where calculated using the program SCÅTTER for three independent HUαα-80bp DNA and HUα$^{E34K}$-80bp DNA titration experiments. Pair distribution function $P(r)$ was computed

using the program GNOM[66]. The distance $r$ where $P(r)$ functions approach zero intensity specifies the maximal dimension (Dmax) of the macromolecule (Supplementary Figs. 2e, f and 3a). $P(r)$ functions were normalized to volume of correlation of the assemblies calculated in SCÅTTER and listed in the Supplementary Table 2.

**Solution structure modeling**. Experimental scattering profiles of HUαα-80bp DNA complexes that showed diffraction peaks were investigated for the behavior of Intensity in the low $q$-region (low angle). In the double logarithmic representation of the SAXS data, the power decay according to $I \propto q^{-2}$ was evident (Supplementary Figure 2a–d), indicating the presence of lamellar structures for the protein–DNA complexes in solution. Lamellar structures were fitted against the lamellar stack Caille function model in SasView (http://www.sasview.org/). The lamellar stack Caille function is defined as random lamellar head/tail/tail/head sheet with Caille structure factor. The scattering intensity $I(q)$ is

$$I(q) = 2\pi \frac{P(q)S(q)}{q^2\delta} \qquad (2)$$

where $P(q)$ is the form factor, $S(q)$ is the structure factor dependent on Caille constant and $\delta$ is the total layer thickness. The combination of number of lamellar plates, Caille parameter (<0.8) and polydispersity in the bilayer thickness were determinants of acceptable fit to the individual scattering curves (Supplementary Table 3). Experimental scattering profiles of HUαα, HUαα-80bp DNA, HUα$^{E34-}$K$_\alpha$$^{E34K}$, and HUα$^{E34K}$α$^{E34K}$-80 bp DNA complexes that did not show diffraction peaks showed a power decay according to $I \propto q^{-1}$ in the double logarithmic representation of the SAXS data (Supplementary Figure 2a–d), indicating the presence of filament-like structures. These data were fitted using atomistic models. We generated a pool of atomistic models of HUαα-80 bp DNA and HUα$^{E34-}$K$_\alpha$$^{E34K}$-80bp DNA with different protein/DNA ratios based on the HUαα-DNA crystal structure at pH 4.5 and HUα$^{E34K}$α$^{E34K}$-DNA, respectively. We utilized FoXS[67] to guide the selection of single representative state that fit the experimental SAXS profiles of HUαα-80 bp DNA complexes that did not show diffraction peaks and HUα$^{E34K}$α$^{E34K}$-80 bp DNA complexes at pH 6.5 and above from their corresponding pools of models. We applied ensemble analysis[68] to match the SAXS curves of HUα$^{E34K}$α$^{E34K}$-80bp DNA complexes at different protein:DNA ratios using various protein–DNA-binding modes in these multi-component systems. A pool of HUα$^{E34K}$α$^{E34K}$-80 bp atomistic models with different protein–DNA ratios and protein–DNA bridging (covering two to three neighboring DNAs) were generated based on our crystals structures. The extended arms of HU and missing regions if any were built based on the complete protein models from HUαα-DNA crystal structure at pH 4.5. A selection of sub-ensemble of HUα$^{E34K}$α$^{E34K}$-80 bp DNA complexes coexisting in solution was guided by the fit to the experimental SAXS data using FoXS and Multi-FoXS[67,69]. The complexes were weighted and allowed selection of a minimal ensemble to avoid over-fitting of SAXS data. The match of theoretical weighted profiles to the experimental curve confirmed a filament-like structure for HUα$^{E34K}$α$^{E34K}$-80bp DNA complexes. Structures were visualized in CHIMERA[53]. Data and the related models were deposited in the SASBDB data base (https://www.sasbdb.org/). The SASBDB data base accession codes and experimental SAXS parameters are reported in Supplementary Table 2.

**Statistics and reproducibility**. Descriptive statistics to obtain mean and standard deviation (for Fig. 1 and Supplementary Figure 3) were performed in OriginLab (OriginLab, Northampton, MA). Statistical significance was also assessed in OriginLab using a two-sided two-sample $t$ test. For cell growth assay, the data were derived from $n = 3$ biological replicates. We used three independent biological replicates for each condition and/or genotype to detect differential gene expression in RNA-sequencing experiments. Statistics for LAC Histogram were determined from $n = 10$ independent bacterial cell for each experimental condition. SAXS experiments were repeated 2/3 times independently. We confirm that all attempts to replicate experiments were successful.

**Reporting summary**. Further information on research design is available in the Nature Research Reporting Summary linked to this Article.

## Data availability

Atomic coordinates and structure factors for the crystal structures are deposited in the Protein Data Bank (PDB–(https://www.rcsb.org/)) PDBID: 6O8Q, 6O6K, and 6OAJ (Fig. 5 and Supplementary Figures 4, 5). SAXS data are deposited in the Small-Angle Scattering Biological Data Bank (SASBDB—and SASDBD IDs are listed in the Supplementary Table 2 and figure legend of Supplementary Figure 3 (Fig. 4 and Supplementary Figures 2, 3). Soft X-ray tomography data are available here (https://ncxt-nas1.lbl.gov:5001/fsdownload/XKnLbGVgY/Nature%20comm). The RNA-Seq are deposited in the Gene Expression Omnibus data base (GEO) with accession code GSE134667 (Fig. 6). All data are available from the authors upon reasonable request.

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

## Acknowledgements

This work was supported in part by the DOE BER Integrated Diffraction Analysis Technologies (IDAT) program and NIGMS grant P30 GM124169 and Advanced Light Source (ALS) resource ALS-ENABLE. SAXS at ALS SIBYLS beamline is also supported by NIH grant CA92584. The National Center for X-ray Tomography is supported by NIH (P41GM103445) and DOE BER (DE-AC02-5CH11231). Beamline 8.3.1 at the Advanced Light Source is operated by the University of California Office of the President, Multicampus Research Programs, and Initiatives grant MR-15-328599 the National Institutes of Health (R01 GM124149 and P30 GM124169), Plexxikon Inc. and the Integrated Diffraction Analysis Technologies program of the US Department of Energy Office of Biological and Environmental Research. This work was partially supported by the gift funds from IUCr2017–Local Organizing Committee, to Dr. Soumya Govinda Remesh winner of the Dragons's Den contest at the IUCr2017 Congress, Hyderabad, India. We thank members of the SIBYLS group (BL 12.3.1), BL 2.1, and BL 8.3.1 at ALS for aiding data collection and for comments and suggestions in preparation of this paper. We thank Andrei Trostel for help in RNA extraction from some of the samples tested here.

## Author contributions

Conceptualization, M.H. and S.A.; methodology (MX, SAXS, SXT) S.G.R. and M.H.; methodology (SXT), J.H.C. and A.E.; validation (MX, SAXS, SXT), S.G.R. and M.H.; validation (RNA-Seq), S.V; formal analysis (MX, SAXS, SXT), S.G.R. and M.H.; formal analysis (RNA-Seq), S.V.; investigation (Protein and DNA purification, crystallization) S.G.R.; investigation (MX, SAXS), S.G.R. and M.H.; investigation (SXT), J.H.C. and S.G.R.; investigation (RNA-Seq), S.V.; writing—original draft, S.G.R. and M.H.; visualization, S.G.R. and M.H.; supervision; S.A, C.L., and M.H.; project administration, S.A., C.L., and M.H.; funding acquisition, S.A. and M.H.

## Competing interests

The authors declare no competing interests.
