## [Peer Review File · Nature Communications]

Reviewers' comments:

Reviewer #1 (Remarks to the Author):

Remesh et al. studied the role of histone-like proteins (HU) in the 3D organization of the bacteria DNA. Specifically, the authors compared the nucleoid architecture of bacteria in growth phase between wild type and HU mutants under different environmental conditions (i.e. at different media pH). For their structural analysis the authors combined two complementary approaches, soft X-ray tomography (SXT) and X-ray crystallography. The former provides mesoscale biophysical and architectural insight to nucleoid organization, allowing molecular predictions that were confirmed at the atomic scale by crystallography. Intriguingly, the authors showed that under stress, the organization of the E. coli genome is remarkably different between the genotypes that they analyzed suggesting an important role of HUs in remodeling the nuclei core upon stress and maintain long range DNA-DNA interactions. The authors showed how the HU α complex only at low pH, self assemble into ordered lamellar structure under physiological conditions. This property is lost when the pH is brought back to neutral. The authors with ad hoc mutation in the HU α showed how HU α -DNA interactions depend on surface electrostatic interaction. Most importantly, they validated their hypothesis and gained mechanistic insights solving the structure of the complex HU α -DNA. Importantly, these structural perturbations are likely to have direct regulatory consequences as nucleoid remodeling by different mutations correlates with variation in gene expression.

I find the combination of SXT, X-ray crystallography, and bacterial genetics powerful and insightful for the understanding of nucleoid organization and its role in bacterial gene regulation. Therefore I recommend publication without additional experiments.

Reviewer #2 (Remarks to the Author):

The paper by Remesh et al. describes the multiple DNA interaction modes exhibited by the bacterial, histone-like protein HU (HU α), and their effects on the bacterial genome architecture and the global gene expression. The authors analyze the HU α E34K mutant, which they previously identified as a mutant that is defective in the multimerization of HU α -HU α dimers (denoted as HU $\alpha\alpha$), throughout the paper, using various techniques. Using soft X-ray tomography, they show that HU α is involved in the formation of a condensed 'core' in the inner region of the nucleoid. Using small-angle X-ray scattering (SAXS) combined with extensive comparison between the experimental and theoretical SAXS profiles, the authors demonstrate that HU $\alpha\alpha$ have several DNA interaction modes that is affected by ionic strength and pH. Using X-ray crystallography, they determined structures of the HU $\alpha\alpha$ -DNA complexes, and revealed the detailed interactions occurring at the HU $\alpha\alpha$ -HU $\alpha\alpha$ interface. Finally, using next-generation RNA sequencing, the authors show some differences in gene expression, in the absence of HU.

The SAXS and X-ray crystallography data correlate very well with each other, and appear consistent with the soft X-ray tomography analysis of the multimerization mutant (HU α E34K) that is defective in forming a condensed nucleoid region. I am basically positive about publishing this work, although some rewriting may be needed to improve this paper.

Specific points:

1) I think Figure 1 is difficult to comprehend. A simple comparison between WT and HU α E34K mutant cells in various growth phases (not including the WT cells under acid stress) would be better. The images of WT cells under acid stress may be shown in a different figure, or in the Supplementary section. Images of the WT and HU α E34K mutant cells in various growth phases should be aligned side-by-side, with the WT cells shown on the left side of the figure. Accordingly, the authors may wish to move the description on the WT cells under acid stress to the latter part

of the Results section (pp. 3-6).

2) Similar to my first point, I recommend moving Fig. 2b to another figure (perhaps combining with the acid stress data in Fig. 1b).

3) Fig. 1a

For consistency with the LAC histograms in Fig. 1b and 1c, the LAC values should increase from left to right.

4) Results section (p. 3), line 3

The HU α E34K mutant is mentioned with no background information. The information that presented in p. 6 (2nd paragraph) should be moved to p. 3.

5) Results section (p. 8), line 11

"Fig. 4a" should be "Fig. 3a".

6) Results section (p. 8), line 15

"increase" should be "decrease".

7) Results section (p. 8), line 8

"The sharpness of the peak" may be simply "A peak", to avoid the contradiction that lamellar assemblies are also suggested in the presence of 100 mM or 50 mM NaCl, which are not sharp peaks.

8) Fig. 3a

Please include the rotation axis.

9) Results section (p. 10), line 4

"over the entire pH range" appears incorrect. Please correct.

10) Results section (p. 10), "Molecular switch in HU α coupling alters DNA bundling"

In addition to pH, it may also be important to mention that sample preparation and crystallization was performed at low ionic strengths.

11) Fig. 4a and 4b

Please include the rotation axis.

Reviewer #3 (Remarks to the Author):

In this study, Remesh et al. utilize a wide array of techniques (soft X-ray tomography, small angle x-ray scattering, crystallography, and RNA-seq) to determine the interplay between HU α charge state and DNA organization/bundling. Through SAXS measurements, they found that under "high" ionic conditions (300 mM), HU forms a filament arrangement along the DNA, but at physiological salt and low pH conditions (150 mM, pH of 4.5 or 5.5), the HU-DNA complexes natively formed in to lamellar structures, with varying characteristic spacings in accordance with pH condition. Lamellar formation was unobserved in the pH 7.5 and E34K-mutant systems, regardless of salt concentration. Analysis of the asymmetric cell from crystallography measurements reinforced the presence of these characteristic spacings according to pH (60 Å for pH 4.5, 70 Å for pH 5.5). RNA-seq measurements showed variations in gene regulation that correlate nicely with observed changes in HU-DNA structure and nucleoid compaction, as measured by x-ray tomography. Altogether, these combination of results provide a cohesive explanation for the potential structure-function role of HU in regulating gene expression in microbes.

That being said, I believe that there remains a few areas of the manuscript that need clarification and/or improvement before the manuscript is ready for publication, mostly relating to reporting of the SAXS modeling:

1) On a number of occasions, the authors state that their models "agree well with experimental SAXS data". However, the only evidence that they provide of such fit qualities are overlays between the modeled and measured SAXS profiles. While most of these profiles appear to fit quite nicely upon visual inspection, fit qualities, such as χ^2 , should be explicitly reported (or perhaps reported more clearly, if I may have missed them in my multiple readings of the manuscript).

2) Additionally, the filament/lamellar structure fits were conducted using FoXS/multiFoXS, according to table S2. However, the number of states predicted in the ensemble is never discussed - the only mention being that multiFoXS's algorithm selects for a minimal ensemble that resists overfitting (paraphrase of author's words). Since these details are not provided, it's difficult to determine if a multi-state ensemble is even necessary to fit the data, or if a singular representative conformation is sufficient (as seems to be suggested by the main text). To this end, I believe that including details regarding the multi-state ensemble of structures predicted from multiFoXS may clarify this issue. Additionally, a brief discussion regarding the fit quality from each sub-ensemble vs the total set of multiple states may also provide information regarding the degeneracy of DNA binding states in each form, in addition to being the proper statistical reporting (in my opinion).

3) On page 13, under the section "HU-dependent nucleoid remodeling correlates to global gene expression changes", the authors state (emphasis added): "we found that 199 genes were up-regulated while **75** genes were down-regulated in the lag-phase" but their numbers printed in figure 5a suggest an up-regulation of 199 genes and a down-regulation of **73** genes, according to the inlayed text. From my reading of the Excel sheet containing Table S3, 73 is indeed the proper value. Similarly, when I inspect the table S5 Excel sheet, I read 75 and 173 significant down- and up-regulated genes, respectively, but the inlay of Figure 5c states 73 and 175. While the discrepancy is minor, the authors should take great care to make sure that these values are consistent.

4) On a minor note, there are multiple points of grammar errors that should be carefully cleaned, most notably proper pluralizations. Additionally, the following repeated statement should be removed (page 8, final sentence, emphasis added): "...dictates the hydrogen bond network **critical to critical to** HUaa...".

On the whole, the manuscript is a rather complete study, considering multiple approaches to understand the structure-function relationship of HU-DNA complexes in regulating transcription. Even researchers not directly interested in HU biochemistry can benefit from this paper, as it provides a good example for combining several different structural techniques to support the paper's conclusions. Additionally, it provides interesting information regarding the role of electrostatics and environmental pH in driving functionality, but the previously listed notes should be addressed before the manuscript is published. However, I feel the necessary modifications to satisfy these comments should be quite minor.

We thank the reviewers for their expert and thorough consideration of our manuscript as well as for providing comments that lead to an improved presentation of our results. All three reviewers noted important new insights into the HU α controlled DNA bundling from our hybrid structural approach (combining X-ray crystallography, solution X-ray scattering and Soft X-ray tomography), along with the general interest and new insight into the regulatory systems of bacterial chromatin remodeling.

Reviewers' comments:

Reviewer #1 (Remarks to the Author):

Remesh et al. studied the role of histone-like proteins (HU) in the 3D organization of the bacteria DNA. Specifically, the authors compared the nucleoid architecture of bacteria in growth phase between wild type and HU mutants under different environmental conditions (i.e. at different media pH). For their structural analysis the authors combined two complementary approaches, soft X-ray tomography (SXT) and X-ray crystallography. The former provides mesoscale biophysical and architectural insight to nucleoid organization, allowing molecular predictions that were confirmed at the atomic scale by crystallography. Intriguingly, the authors showed that under stress, the organization of the E. coli genome is remarkably different between the genotypes that they analyzed suggesting an important role of HUs in remodeling the nuclei core upon stress and maintain long range DNA-DNA interactions. The authors showed how the HU α complex only at low pH, self assemble into ordered lamellar structure under physiological conditions. This property is lost when the pH is brought back to neutral. The authors with ad hoc mutation in the HU α showed how HU α -DNA interactions depend on surface electrostatic interaction. Most importantly, they validated their hypothesis and gained mechanistic insights solving the structure of the complex HU α -DNA. Importantly, these structural perturbations are likely to have direct regulatory consequences as nucleoid remodeling by different mutations correlates with variation in gene expression.

I find the combination of SXT, X-ray crystallography, and bacterial genetics powerful and insightful for the understanding of nucleoid organization and its role in bacterial gene regulation. Therefore, I recommend publication without additional experiments.

We thank the reviewer for such positive feedback. It is very encouraging and we hope this multi-scale imaging approach will become more mainstream in the future.

Reviewer #2 (Remarks to the Author):

The paper by Remesh et al. describes the multiple DNA interaction modes exhibited by the bacterial, histone-like protein HU (HU α), and their effects on the bacterial genome architecture and the global gene expression. The authors analyze the HU α E34K mutant, which they previously identified as a mutant that is defective in the multimerization of HU α -HU α dimers (denoted as HU $\alpha\alpha$), throughout the paper, using various techniques. Using soft X-ray tomography, they show that HU α is involved in the formation of a condensed 'core' in the inner region of the nucleoid. Using small-angle X-ray scattering (SAXS) combined with extensive comparison between the experimental and theoretical SAXS profiles, the authors demonstrate that HU $\alpha\alpha$ have several DNA interaction modes that is affected by ionic strength and pH. Using X-ray crystallography, they determined structures of the HU $\alpha\alpha$ -DNA complexes, and revealed the detailed interactions occurring at the HU $\alpha\alpha$ -HU $\alpha\alpha$ interface. Finally, using next-generation RNA sequencing, the authors show some differences in gene expression, in the absence of HU.

The SAXS and X-ray crystallography data correlate very well with each other, and appear consistent with the soft X-ray tomography analysis of the multimerization mutant (HU α E34K) that is defective in forming a condensed nucleoid region. I am basically positive about publishing this work, although some rewriting may be needed to improve this paper.

We thank the reviewer for the positive feedback and for the valuable comments. Below we have addressed the specific points highlighted by the reviewer to improve the manuscript.

Specific points:

1) I think Figure 1 is difficult to comprehend. A simple comparison between WT and HU α E34K mutant cells in various growth phases (not including the WT cells under acid stress) would be better. The images of WT cells under acid stress may be shown in a different figure, or in the Supplementary section. Images of the WT and HU α E34K mutant cells in various growth phases should be aligned side-by-side, with the WT cells shown on the left side of the figure. Accordingly, the authors may wish to move the description on the WT cells under acid stress to the latter part of the Results section (pp. 3-6).

We have changed Figure 1 as suggested by the reviewer to allow easy comparison between WT and HU α E34K mutant cells. We retain WT cells under acid stress orthoslices in Figure 1. We have moved Figure 1a to Figure 2a.

Our initial observations about pH related changes in nucleoid architecture prompted us to evaluate the effect of mutation of an ionized residue critical to HU-HU multimerization on the nucleoid architecture. Thus, we maintain the results section as is but have made changes to Figure 1.

2) Similar to my first point, I recommend moving Fig. 2b to another figure (perhaps combining with the acid stress data in Fig. 1b).

We have changed Figure 2 to now only include the 3D reconstructions of wild type E.coli cells in different growth phases under normal growth conditions (Figure 2b). Figure 2a (previously Fig 1a) serves to introduce the readers to soft X-ray tomography 3D volumetric reconstructions. We have added new Figure 3 in which we display the 3D reconstructions depicting bacterial nucleoid remodeling under acidic stress and upon HU α mutation with direct comparison to control cells.

3)Fig.1a

For consistency with the LAC histograms in Fig. 1b and 1c, the LAC values should increase from left to right.

We have changed the order of LAC values. Figure 1a has been moved to Figure 2a.

4) Results section (p. 3), line 3

The HU α E34K mutant is mentioned with no background information. The information that presented in p. 6 (2nd paragraph) should be moved to p. 3.

We have added background information of the HU α E34K mutant cells in p. 3. Addition to the text reads -

“Glu34 was previously identified to be critical to hydrogen bond formation for HU α -HU α multimerization that maintains HU α /DNA nucleoprotein complexes in-vitro⁸.”

5) Results section (p. 8), line 11

"Fig. 4a" should be "Fig. 3a".

We thank the reviewer for catching this. We have corrected the error. Figure 3a is now Figure 4a

6) Results section (p. 8), line 15

"increase" should be "decrease".

We thank the reviewer for catching this. We have corrected the error.

7) Results section (p. 8), line 8

"The sharpness of the peak" may be simply "A peak", to avoid the contradiction that lamellar assemblies are also suggested in the presence of 100 mM or 50 mM NaCl, which are not sharp peaks.

We have changed "The sharpness of the peak" to "peak".

8) Fig. 3a

Please include the rotation axis.

We have added rotation axes to Figure 3a (now Figure 4a).

9) Results section (p. 10), line 4

"over the entire pH range" appears incorrect. Please correct.

We have edited the sentence. The sentence now reads –

“We found that the positively charged HU α ^{E34K} mutation disrupted DNA bundling in a protein concentration-independent manner and led to the formation of filament-like nucleoprotein structures **at pH 6.5 , 7.5 and aggregations at pH 5.5, 4.5 with no diffraction peaks** (Fig. 4f).”

10) Results section (p. 10), "Molecular switch in HU α coupling alters DNA bundling"

In addition to pH, it may also be important to mention that sample preparation and crystallization was performed at low ionic strengths.

We have edited the sentence to include information of the ionic strength at which crystallization experiments were performed. The sentence now reads –

“In order to elucidate the molecular-level mechanism that alters the DNA bundling, we determined crystal structures of HU α in complex with 19 bp DNA of random sequence at pH ~4.5 and ~5.5 **and low ionic strength (~ 100-150 mM NaCl)**”

11) Fig. 4a and 4b

Please include the rotation axis.

We have added rotation axes to Figure 4a,b (now Figure 5a,b)

Reviewer #3 (Remarks to the Author):

In this study, Remesh et al. utilize a wide array of techniques (soft X-ray tomography, small angle x-ray scattering, crystallography, and RNA-seq) to determine the interplay between HU α charge state and DNA organization/bundling. Through SAXS measurements, they found that under "high" ionic conditions (300 mM), HU forms a filament arrangement along the DNA, but at physiological salt and low pH conditions (150 mM, pH of 4.5 or 5.5), the HU-DNA complexes natively formed in to lamellar structures, with varying characteristic spacings in accordance with pH condition. Lamellar formation was unobserved in the pH 7.5 and E34K-mutant systems, regardless of salt concentration. Analysis of the asymmetric cell from crystallography measurements reinforced the presence of these characteristic spacings according to pH (60 Å for pH 4.5, 70 Å for pH 5.5). RNA-seq measurements showed variations in gene regulation that correlate nicely with observed changes in HU-DNA structure and nucleoid compaction, as measured by x-ray tomography. Altogether, these combinations of results provide a cohesive explanation for the potential structure-function role of HU in regulating gene expression in microbes.

Thank you.

That being said, I believe that there remains a few areas of the manuscript that need clarification and/or improvement before the manuscript is ready for publication, mostly relating to reporting of the SAXS modeling:

We thank the reviewer for a positive response and for the insightful comments. We provide a point-by-point response to the reviewer's comments below.

1) On a number of occasions, the authors state that their models "agree well with experimental SAXS data". However, the only evidence that they provide of such fit qualities are overlays between the modeled and measured SAXS profiles. While most of these profiles appear to fit quite nicely upon visual inspection, fit qualities, such as χ^2 , should be explicitly reported (or perhaps reported more clearly, if I may have missed them in my multiple readings of the manuscript).

We have added χ^2 (χ^2) values to Supplementary Table 2. We have also added χ^2 values to Supplementary Figure 3. We have corrected the table to reflect only the rigid body fitting protocol used for individual curves. While the main role of SAXS study presented here is to distinguish formation of lamellar and filament structures, we wish to keep details about quality of the SAXS fit for the filament structures in the supplementary information.

2) Additionally, the filament/lamellar structure fits were conducted using FoXS/multiFoXS, according to table S2. However, the number of states predicted in the ensemble is never discussed - the only mention being that multiFoXS's algorithm selects for a minimal ensemble that resists overfitting (paraphrase of author's words). Since these details are not provided, it's difficult to determine if a multi-state ensemble is even necessary to fit the data, or if a singular representative conformation is sufficient (as seems to be suggested by the main text). To this end, I believe that including details regarding the multi-state ensemble of structures predicted from multiFoXS may clarify this issue. Additionally, a brief discussion regarding the fit quality from each sub-ensemble vs the total set of multiple states may also provide information regarding the degeneracy of DNA binding states in each form, in addition to being the proper statistical reporting (in my opinion).

We sincerely apologize for any confusions that Supplementary Table 2 may have led to. We have corrected the table to reflect only the rigid body fitting protocol used for individual curves.

We utilized theoretical lamellar model functions in SASView to fit SAXS data of protein/DNA complexes that showed diffraction peaks. On the other hand we fit atomistic models of HU-DNA filaments to the SAXS data that follow power decay according to $I \propto q^{-1}$ and missing diffraction peak with FoXS. We used FoXS to fit the SAXS data of the protein/DNA complexes at higher pH and ionic strength (pH 5.5, 300 mM NaCl; pH 6.5, 100 mM NaCl and above and pH 7.5, all salt concentrations) against filament structures as shown in the cartoon in Figure 4c,d. We also used FoXS to fit SAXS data of HU α E34K mutant/DNA complexes at pH 6.5 and above against filament structures as shown in Figure 4e. Although, additional filament models could be present in solution we wanted to refrain from over-fitting the data yet highlight the changes in the solution state of the nucleoprotein complexes going from lamellar to filament structures. Aggregation-free SAXS profiles of the concentration series of HU α E34K mutant/DNA complexes (Supplementary Figure 3) were amenable to further analyses using FoXS/MultiFoXS. We show the multi-state ensemble for concentration series of HU α E34K mutant/DNA complexes in Supplementary Figure 3a. We have added panels that show the difference in χ^2 for best-fit (single-state) versus multi-state ensemble in Supplementary Figure 3a. χ^2 values for multi-state ensemble for the concentration series of HU α E34K mutant/DNA complexes are now added to the figure legends of Supplementary Figure 3. We have also edited the 'Methods' section for clarity. We also deposited all SAXS data in SASBDB databank which will become available upon publication. For SASDBD ID # please see Supplementary Table 2.

3) On page 13, under the section "HU-dependent nucleoid remodeling correlates to global gene expression changes", the authors state (emphasis added): "we found that 199 genes were up-regulated while ****75**** genes were down-regulated in the lag-phase" but their numbers printed in figure 5a suggest an up-regulation of 199 genes and a down-regulation of ****73**** genes, according to the inlayed text. From my reading of the Excel sheet containing Table S3, 73 is indeed the proper value. Similarly, when I inspect the table S5 Excel sheet, I read 75 and 173 significant down- and up-regulated genes, respectively, but the inlay of Figure 5c states 73 and 175. While the discrepancy is minor, the authors should take great care to make sure that these values are consistent.

We thank the reviewer for this thorough reading of the manuscript and agree that discrepancies pointed out by the reviewer can cause unintended confusion. We have thus corrected Figure 5a,b (now Figure 6a,b) to reflect these changes and have made the necessary edits to the manuscript as well.

4) On a minor note, there are multiple points of grammar errors that should be carefully cleaned, most notably proper pluralizations. Additionally, the following repeated statement should be removed (page 8, final sentence, emphasis added): "...dictates the hydrogen bond network *****critical to critical to***** HU α ...".

We carefully correct all obvious grammar errors.

On the whole, the manuscript is a rather complete study, considering multiple approaches to understand the structure-function relationship of HU-DNA complexes in regulating transcription. Even researchers not directly interested in HU biochemistry can benefit from this paper, as it provides a good example for combining several different structural techniques to support the paper's conclusions.

Thank you!

Additionally, it provides interesting information regarding the role of electrostatics and environmental pH in driving functionality, but the previously listed notes should be addressed before the manuscript is published. However, I feel the necessary modifications to satisfy these comments should be quite minor.

REVIEWERS' COMMENTS:

Reviewer #1 (Remarks to the Author):

Authors responded to initial critiques in a satisfactory fashion and I am in favor of publication as is.

Reviewer #2 (Remarks to the Author):

The authors have effectively addressed the issues raised in the previous review. I recommend publication of the manuscript.

Wataru Kagawa

Reviewer #3 (Remarks to the Author):

The authors have addressed my SAXS-related criticisms with satisfactory changes to the manuscript text, well designed figure edits, and a thorough rebuttal regarding the choices of states and modelling procedure. I also appreciate their deposition of data in to the SASDBD, in the spirit of open data transparency.

One small point of discrepancy that still remains is the values quoted in (what is now) Figure 6b and Table S5 (differential expression in $\Delta hupA$ $\Delta hupB$). Table S5 still looks to have 75 entries with negative $\log(\text{Fold Change})$ and 173 values of positive $\log(\text{Fold Change})$. Again, I want to stress that the difference in these values (173 in the Excel sheet vs 175 in the figure panel) certainly does not hinder their interpretation, but consistent reporting of values is important for the field, in general.

I believe that the manuscript should be approved for publication, pending the very minor edit of panel 2 in Figure 6b so that it appropriately agrees with Table S5.

REVIEWERS' COMMENTS:

Reviewer #1 (Remarks to the Author):

Authors responded to initial critiques in a satisfactory fashion and I am in favor of publication as is. We thank the reviewer for the favorable recommendation for publication.

Reviewer #2 (Remarks to the Author):

The authors have effectively addressed the issues raised in the previous review. I recommend publication of the manuscript.

Wataru Kagawa

We are pleased that the reviewer is satisfied with the revised manuscript. We thank the reviewer for the recommendation.

Reviewer #3 (Remarks to the Author):

The authors have addressed my SAXS-related criticisms with satisfactory changes to the manuscript text, well designed figure edits, and a thorough rebuttal regarding the choices of states and modelling procedure. I also appreciate their deposition of data in to the SASDBD, in the spirit of open data transparency.

One small point of discrepancy that still remains is the values quoted in (what is now) Figure 6b and Table S5 (differential expression in $\Delta hupA$ $\Delta hupB$). Table S5 still looks to have 75 entries with negative $\log(\text{Fold Change})$ and 173 values of positive $\log(\text{Fold Change})$. Again, I want to stress that the difference in these values (173 in the Excel sheet vs 175 in the figure panel) certainly does not hinder their interpretation, but consistent reporting of values is important for the field, in general.

I believe that the manuscript should be approved for publication, pending the very minor edit of panel 2 in Figure 6b so that it appropriately agrees with Table S5.

We are pleased that the revised manuscript particularly addressing the SAXS-related criticisms are viewed as satisfactory by the reviewer. We would sincerely like to apologize a second time for the discrepancy between the Figure 6a and Supplementary Table 5. We would like to assure the reviewer that it has been corrected in the final version.